# RETHINKING THE IDENTIFICATION CAPABILITY OF OUT-OF-DISTRIBUTION DETECTION

## ABSTRACT

Out-of-distribution (OOD) detection aims to identify semantically shifted data, i.e., samples outside the categories of in-distribution (ID) training data. However, prior studies primarily focus on detecting perfect ID and OOD data, where they exhibit significant covariate shift, i.e., differing distributions of input images with the same semantic label, overlooking the imperfect data setup in which the covariate shift may be negligible. Consequently, a significant covariate shift between ID and OOD samples can serve as a shortcut for OOD detectors, leading to doubts about whether existing OOD detectors truly identify semantic shifts or merely exhibit good performance due to the covariate shift. In response to such doubts, we conduct a theoretical analysis that demonstrates the learnability of the OOD detection task degrades with decreasing covariate shift. In this paper, we investigate a novel and challenging problem termed **C**ovariate-**S**hift-**F**ree **S**etting (CSFS), aiming to perform OOD detection in the case of ruling out the impact of covariate shift. To address the CSFS, we propose a novel approach that leverages class-specific gradients as an efficient signal to disentangle OOD features from imperfect ID data for model fine-tuning. Extensive experiments show that our method, using only imperfect ID data, outperforms all existing counterparts, including those employing additional OOD data for model training, across various OOD detection setups.

## 1 INTRODUCTION

To prevent machine learning (ML) models from arbitrarily classifying out-of-distribution (OOD) data as in-distribution (ID) data, extensive research has been devoted to the area of OOD detection, which plays a crucial role in safety-critical domains such as autonomous driving, medical diagnosis, and financial security (Wang et al., 2023a; Bai et al., 2023; Du et al., 2024). To be specific, current research either focuses on designing new OOD scoring functions based on fixed models (Hendrycks & Gimpel, 2017; Lee et al., 2018c; Liu et al., 2020; Sun et al., 2022), or developing efficient OOD fine-tuning strategies (Du et al., 2022; Lee et al., 2018a; Tack et al., 2020; Hendrycks et al., 2019), which has yielded remarkable progress in detecting OOD samples.

In principle, OOD detection aims to identify OOD samples with semantic shift, i.e., owning *different semantic labels* from the given ID classes (Yang et al., 2024; Fang et al., 2022; Du et al., 2024). However, we observe that an implicit assumption is commonly embodied in previous OOD detection methods: OOD samples exhibit not only semantic shift (different class labels) but also *covariate shift (different distributions on input images with the same class label, e.g., divergent styles)* from ID samples. Consequently, an intuitive question arises, i.e., whether the covariate shift exhibited in previous work offers a shortcut to the goal of OOD detection? In other words, the potentially large might serve as an auxiliary, powerful signal for an OOD detector, while the underlying semantic shift is overlooked (the utimate goal of OOD detection).

In response to such doubt, we aim to investigate a novel but challenging problem, termed **C**ovariate-**S**hift-**F**ree **S**etting (CSFS), aiming to identify OOD samples by ruling out the impact of covariate shift between OOD and ID data. We investigate CSFS by assessing the OOD detectors in the setup shown in Fig. 1, where ID and OOD features are obtained from the same data acquisition process, resulting in negligible covariate shift between them. This differs from prior work that assumes OOD detection with shortcut, i.e., potentially significant covariate shifts between ID and OOD features.

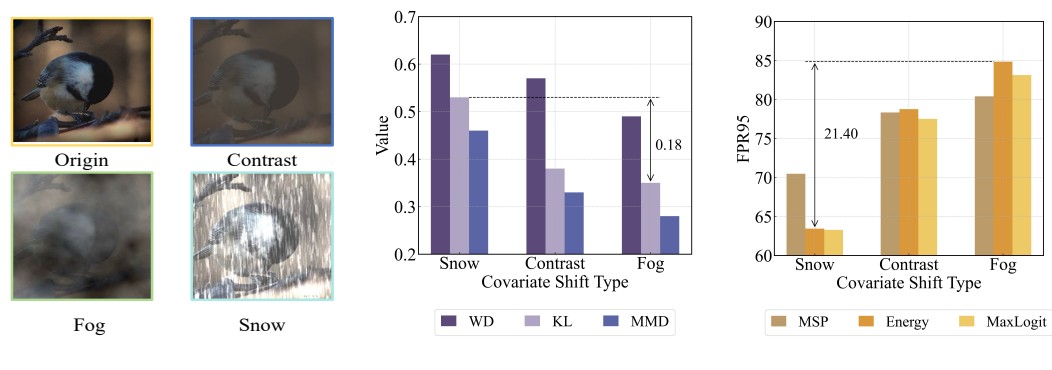

(a) Covariate Shifted Samples     (b) Distribution Distance     (c) OOD Detection Performance

Figure 1: Fig. 1(a) illustrates examples of covariate shift. Fig. 1(b) illustrates the distribution distances between CIFAR-100 and three different corrupted versions of ImageNet-200 under Wasserstein distance (WD), KL-divergence (KL), and Maximum Mean Discrepancy (MMD). To enhance visualization, we normalized the distance values to [0,1] while retaining their relative magnitudes. Fig. 1(c) shows the OOD detection performance (CIFAR-100 as ID) evaluated by FPR95 for three classical OOD detection methods. **It is evident that despite the constant semantic shift between ID and OOD features, the performance of OOD detection deteriorates as the covariate shift decreases.**

Moreover, our theoretical analysis informs that an increasing covariate shift significantly improves the Probably Approximately Correct (PAC) learnability of the OOD detection task.

To offer effective OOD detection solutions for our CSFS, one critical challenge remains unexplored: *how to distinguish ID and OOD features with nearly vanishing covariate shift*. To this end, we propose a novel method, termed **D**isentangling **I**mperfect **ID** **D**ata (DIID). Based on the insight that fine-tuning the model with ID and OOD data free of covariate shift can enhance its ability to distinguish them in such scenarios, we employ the class-specific gradient as an important and computationally efficient signal to disentangle covariate-shift-free ID and OOD features from the data for fine-tuning, thereby further improving the discriminative capability of the model. Furthermore, we fine-tune the model to assign higher probabilities to ID data and lower to OOD data, and introduce a regularization term to penalize the model for incorrectly detecting ID instances as OOD.

We conduct extensive experiments across various OOD detection settings, demonstrating the superior performance of our method in practical applications. For instance, in the CSFS setting with ViT-B/16 (Dosovitskiy et al., 2020) as the backbone model, our method achieves a 23.71% reduction in FPR95 (from 28.47% to 4.76%) compared with state-of-the-art (SOTA) methods using only imperfect ID data, and a 16.70% reduction (from 21.46%) compared with SOTA methods that fine-tune the model with additional auxiliary OOD data. Notably, our method achieves these improvements using only imperfect ID data, without any additional auxiliary OOD data. Moreover, it consistently outperforms counterparts in other OOD detection settings, highlighting its reliability and robustness.

The key contributions of this paper are summarized as follows.

- Based on our observation that the performance of OOD detection deteriorates as the covariate shift between ID and OOD features decreases, even when the semantic shift remains constant, we introduce the CSFS problem and conduct a comprehensive theoretical analysis, which demonstrates that as the covariate distribution shift between ID and OOD features diminishes, the PAC learnability of the OOD detection task also diminishes.

- To tackle the CSFS problem, we propose a novel and efficient method DIID, which disentangles ID and OOD features from imperfect ID data using the class-specific gradient as the disentangling signal, and employs the obtained OOD features (without covariate shift against ID features) to fine-tune the model, thereby enhancing its OOD detection performance.

- We evaluate our method across various OOD testing settings, demonstrating superior performance compared to all existing methods. Furthermore, to evaluate performance under CSFS, we construct a new test OOD dataset based on the ImageNet-1k dataset.

## 2 PRELIMINARIES

In this section, we introduce the primary notations used throughout the paper and elaborate on the relevant concepts. Additionally, we detail the problem setting of imperfect ID data.

### 2.1 NOTATIONS

Let $\mathcal{X}$ represent the feature space and $\mathcal{Y} = \{1, 2, \cdots, c\}$ denote the label space corresponding to the ID distribution. We consider the perfect ID feature random variable $\tilde{X}_\mathrm{I}$, imperfect ID feature random variable $X_\mathrm{I}$, and OOD feature random variable $X_\mathrm{O}$, where the realization $\tilde{\mathbf{x}}$ of $\tilde{X}_\mathrm{I}$ contains only ID features, but the realization $\mathbf{x}$ of $X$ contains both ID and OOD features. Accordingly, we have the perfect ID joint feature distribution $D_{\tilde{X}_\mathrm{I} Y_\mathrm{I}}$, imperfect ID joint feature distribution $D_{X_\mathrm{I} Y_\mathrm{I}}$, and OOD feature distribution $D_{X_\mathrm{O} Y_\mathrm{O}}$, where $Y_\mathrm{I}$ is a random variable with outputs in $\mathcal{Y}$ and $Y_\mathrm{O}$ is a random variable whose outputs lie outside $\mathcal{Y}$, i.e., $Y_\mathrm{O} \notin \mathcal{Y}$. Moreover, we also have the perfect marginal ID feature distribution $D_{\tilde{X}_\mathrm{I}}$, imperfect marginal ID feature distribution $D_{X_\mathrm{I}}$, and marginal OOD feature distribution $D_{X_\mathrm{O}}$. Obviously, there exists a non-empty intersection between the supports of $D_{X_\mathrm{I}}$ and $D_{X_\mathrm{O}}$. During testing, we consider a mixture of marginal ID and OOD feature distributions, $D_X^{cov} := (1 - \alpha) D_{X_\mathrm{I}} + \alpha D_{X_\mathrm{O}^{cov}}$, where the constant $\alpha \in [0, 1)$ denotes an unknown class-prior probability (Fang et al., 2022). Here, *cov* denotes covariate shift between ID and OOD features. In this paper, we focus on addressing CSFS, where there is no covariate shift between ID and OOD features, i.e., $D_X^{cov} := (1 - \alpha) D_{X_\mathrm{I}} + \alpha D_{X_\mathrm{O}}$. Furthermore, we consider the classification model $\mathbf{f} : \mathcal{X} \mapsto \mathbb{R}^c$ with logit outputs. In addition, we use 'sof' to denote the softmax operation.

### 2.2 OOD DETECTION

Here, we introduce the formal definition of OOD detection with perfect ID data.

**Goal of OOD Detection** Fang et al. (2022). The goal of OOD detection is to train a predictor $\mathbf{f}$ using an ID training dataset $\tilde{S}_\mathrm{I} := (\tilde{\mathbf{x}}_\mathrm{I}^1, y_\mathrm{I}^1), \cdots, (\tilde{\mathbf{x}}_\mathrm{I}^n, y_\mathrm{I}^n) \overset{\text{i.i.d.}}{\sim} D_{\tilde{X}_\mathrm{I} Y_\mathrm{I}}$. At the testing stage, the predictor is expected to correctly classify a sample $\mathbf{x}$ if $\mathbf{x} \sim D_\mathrm{I}$, and to detect it as an OOD instance if $\mathbf{x} \sim D_\mathrm{O}^{cov}$.

**OOD Scoring.** Based a well-trained model $\mathbf{f}$, the aim of OOD detection at inference time is realized by utilizing various score-based strategies Hendrycks & Gimpel (2017); Lee et al. (2018c); Liu et al. (2020); Sun et al. (2022). Specifically, given an OOD scoring function, denoted as $s(\cdot; \mathbf{f}) : \mathcal{X} \mapsto \mathbb{R}$, the OOD detector $h_\rho$ is given by:

$$h_\rho(\mathbf{x}) = \text{ID, if } h(\mathbf{x}; \mathbf{f}) \geq \rho; \text{ otherwise, } h_\rho(\mathbf{x}) = \text{OOD,}$$

where $\rho$ is a well-defined threshold. We present a widely used example as follows.

**Maximum Softmax Prediction (MSP).** As a well-known baseline, Hendrycks & Gimpel (2017) employs the maximum softmax probability as the OOD scoring, with the scoring function defined as:

$$s_\text{MSP}(\mathbf{x}; \mathbf{f}) = \max_j \text{ sof}_j \mathbf{f}^j(\mathbf{x}), \tag{1}$$

where $\text{softmax}_j(\cdot)$ indicates the $j$-th softmax output. Ideally, ID samples are assigned higher softmax outputs along the dimension corresponding to their true label. Conversely, OOD data, which do not belong to any ID class, should exhibit uniformly low softmax outputs across all dimensions.

### 2.3 CSFS: COVARIATE-SHIFT-FREE SETTING

Although existing research has proposed numerous OOD scoring functions with theoretical guarantees, these methods implicitly assume that there is a substantial covariate shift between ID and OOD features, thereby neglecting the possibility of minimal covariate shift. In order to achieve more robust OOD detection, we explore whether OOD detectors are truly identifying OOD features based on semantic shifts or merely exploiting the shortcuts provided by covariate shifts. We conducted a systematic theoretical analysis and arrived at the conclusions as stated in the following theorem.

**Theorem 1.** *The Probably Approximately Correct (PAC) learnability of the OOD detection task diminishes as the covariate shift between ID and OOD features decreases.*

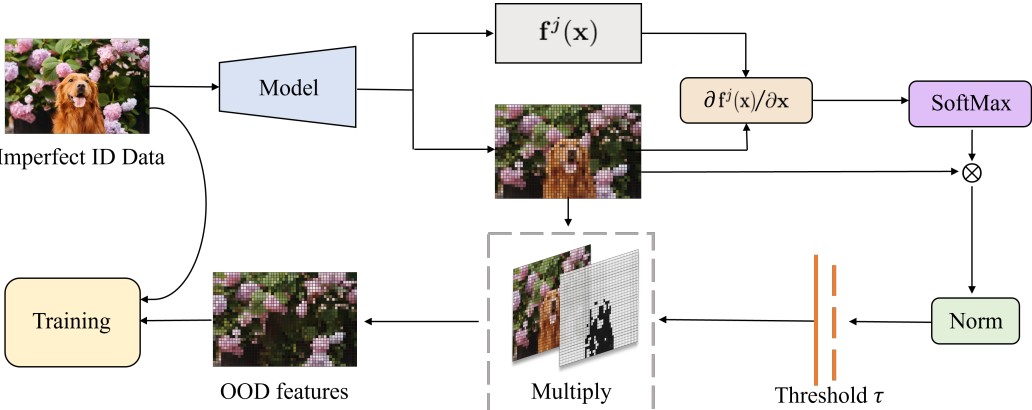

Figure 2: The framework of our proposed method, DIID. We compute class-specific gradients for input $\mathbf{x}$, utilize them to weight the input features, and apply a predefined threshold $\tau$ to filter out portions below the threshold as OOD features that are then employed to fine-tune the model.

*Proof.* Due to space limitations, the detailed theoretical proofs are provided in Appendix B. □

Based on the conclusions derived from Theorem 1, we introduce a novel and challenging OOD detection problem, termed the Covariate-Shift-Free Setting (CSFS), formally defined as follows.

**Problem 1** (Covariate-Shift-Free Setting (CSFS)). *Given an imperfect ID joint feature distribution $D_{X_I Y_I}$ and a training ID dataset $S_I := \{(\mathbf{x}_I^1, y_I^1), \cdots, (\mathbf{x}_I^n, y_I^n)\}$ drawn independent and identically distributed from $D_{X_I Y_I}$, the goal of OOD detection under CSFS is to train a classifier $\mathbf{f}$ by utilizing the training data $S_I$ such that, for any test feature $\mathbf{x}$ drawn from the mixed marginal distribution $D_X$:*

*(a) if $\mathbf{x}$ is an observation from $D_{X_I}$, the trained $\mathbf{f}$ can classify $\mathbf{x}$ into its correct ID label; otherwise (b) if $\mathbf{x}$ is an observation from $D_{X_O}$, the trained $\mathbf{f}$ can detect $\mathbf{x}$ as an OOD case.*

**Remark 1.** *We define CSFS using the imperfect ID joint distribution, since directly collecting data from the real world inevitably results in imperfect ID data. In many large-scale systems, data processing is time-consuming and labor-intensive, and addressing OOD detection with imperfect ID data can substantially reduce cost. Furthermore, CSFS focuses on an OOD marginal feature distribution $D_{X_O}$ that exhibits no covariate shift with respect to the ID marginal feature distribution.*

## 3 METHODOLOGY

In this section, we introduce an efficient approach to address the CSFS problem. With the intuitive insight that fine-tuning the model directly with ID and OOD data free of covariate shift can enhance its ability to distinguish between ID and OOD samples in this setting. To be specific, we disentangle OOD features from imperfect ID data and then fine-tune the model so that it assigns higher prediction probabilities to ID features while reducing the prediction probabilities for these OOD features. Moreover, we penalize the model whenever it incorrectly classifies ID data as OOD during fine-tuning. Then, the first challenge we need to address is *how to effectively disentangle OOD features from imperfect ID data that simultaneously contain both ID and OOD features.*

**Gradient Signals For ID-OOD Feature Disentanglement.** To this end, we observe that neural networks trained via gradient descent become increasingly sensitive to class-relevant features in the input as training progresses. Specifically, when mapping input features to class prediction scores, features associated with the target class contribute most significantly to the output. Consequently, when computing the gradient of the output with respect to the input features, gradients corresponding to class-relevant features are larger than those of class-irrelevant features. Based on this observation, we can utilize class-specific gradients as a signal to distinguish between ID features and OOD features within imperfect ID data. Based on this observation, the method for disentangling OOD features,

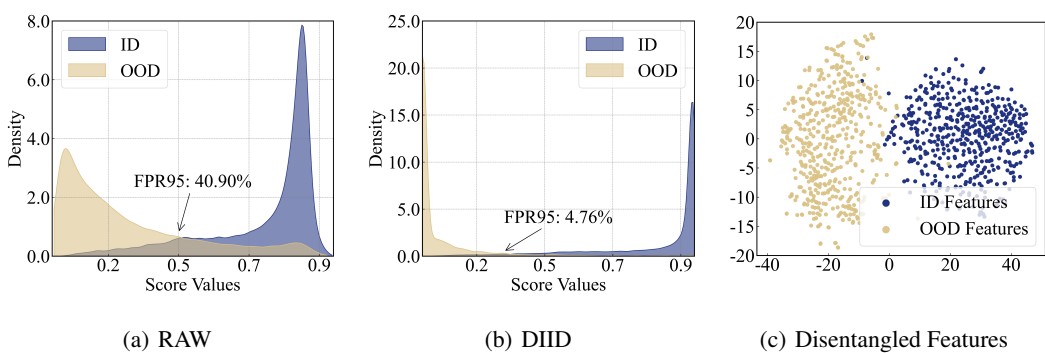

(a) RAW $\qquad$ (b) DIID $\qquad$ (c) Disentangled Features

Figure 3: Figs. 3(a)-3(b) illustrate the distribution of OOD scores before (RAW) and after (DIID) fine-tuning with our DIID, respectively, both employing MSP as the scoring function. After fine-tuning with DIID, the distribution of OOD scores for ID features and OOD features shows a greater separation, making them more distinguishable. Fig. 3(c) presents the t-SNE visualization of ID features and OOD features disentangled from imperfect ID data by DIID.

denoted as $\mathbf{x}_O$ from imperfect ID data $\mathbf{x}$ can be formulated as follows:

$$\mathbf{x}_O = \mathbf{1}\{\arg\max_j \mathbf{f}^j(\mathbf{x}) = y_I\}\mathbf{1}\{\texttt{norm}(\texttt{sof}(\frac{\partial \mathbf{f}^j(\mathbf{x})}{\partial \mathbf{x}}) \cdot \mathbf{x}) \leq \tau\} \cdot \mathbf{x}, \tag{2}$$

where $\texttt{norm}(\cdot)$ represents the normalization operation, and $\tau$ is a pre-defined parameter in the range [0,1]. Specifically, the indicator function $\mathbf{1}\{\arg\max_j \mathbf{f}^j(\mathbf{x}) = y_I\}$ ensures that the model has the capability to correctly classify the input imperfect ID data $\mathbf{x}$. This is crucial because only when the model correctly classifies the input can the gradients of the output predictions with respect to the features serve as a reliable signal. The softmax operation on the gradients amplifies the difference between the gradients of class-relevant features and those of class-irrelevant features. After passing through two indicator functions, a mask matrix is obtained, where the values corresponding to class-irrelevant features are set to 1, and those corresponding to class-relevant features are set to 0. The disentangled OOD features are obtained by multiplying the input features with the mask matrix.

**Learning Strategy.** After disentangling OOD features from the imperfect ID data, we fine-tune the model using these features in an Outlier Exposure (OE)-based manner to enhance its OOD detection capability (Hendrycks et al., 2019). OE treats OOD detection as a binary classification task, leveraging additional auxiliary OOD data to train the model to distinguish between ID and OOD data. The empirical risk of OE can be formulated as follows:

$$\widehat{R}_{OE}(\mathbf{f}) = \widehat{R}_I(\mathbf{f}) + \beta \cdot \widehat{R}_A(\mathbf{f}), \tag{3}$$

Where $\widehat{R}_I(\mathbf{f})$ handles the multi-class classification task for ID data, while $\widehat{R}_A(\mathbf{f})$ handles the binary classification task between ID and OOD data, and $\beta$ is a trade-off parameter. Furthermore, $\widehat{R}_I(\mathbf{f})$ and $\widehat{R}_A(\mathbf{f})$ can be formulated as

$$\widehat{R}_I(\mathbf{f}) = \frac{1}{n}\sum_{i=1}^{n}\ell_I(\mathbf{f}(\mathbf{x}^i), y_I^i) \text{ and } \widehat{R}_A(\mathbf{f}) = \frac{1}{m}\cdot\sum_{t=1}^{m}\ell_{OE}(\mathbf{f}(\mathbf{x}_A^t)). \tag{4}$$

We introduce a novel regularization term that penalizes the network for assigning higher prediction scores to OOD features than to ID features. This regularization term is formulated as follows:

$$\widehat{R}_{reg}(\mathbf{f}) = \frac{1}{nm}\sum_{i=1}^{n}\sum_{t=1}^{m}\left[\mathbf{1}\{\arg\max_j \mathbf{f}^j(\mathbf{x}^i) = y_I^i\}\ell_{reg}(\texttt{sof}(\mathbf{f}(\mathbf{x}^i)) - \texttt{sof}(\mathbf{f}(\mathbf{x}_O^t)))\right]. \tag{5}$$

Then, we fine-tune the model by minimizing the following empirical risk:

$$\widehat{R}(\mathbf{f}) = \widehat{R}_I(\mathbf{f}) + \beta \cdot \widehat{R}_O(\mathbf{f}) + \gamma \cdot \widehat{R}_{reg}(\mathbf{f}), \tag{6}$$

where $\gamma$, similar to $\beta$, is also a trade-off parameter.

The overall framework of our method is illustrated in Fig. 2. Furthermore, we visualize the performance of our method in both disentangling imperfect ID data and fine-tuning the model in Fig. 3, with results obtained using ImageNet-1k as ID data, ImageNet-Real-O as OOD data, and ViT-B/16 as the backbone. Figs. 3(a)-3(b) depict the distribution of OOD scores predicted by the model before and after fine-tuning, respectively. As can be seen, fine-tuning with our method significantly enhances the OOD detection capability of the model, reducing the overlap in predicted OOD scores between ID and OOD features. Specifically, FPR95 decreases from 40.9% to 4.76%. And Fig. 3(c) shows the disentangled ID and OOD features, demonstrating the effectiveness of our method to separate them.

## 4 EXPERIMENTS

In this Section, we conduct extensive experiments to validate the OOD detection performance of our proposed method, DIID. We commence by introduce the experiment setups.

### 4.1 EVALUATION SETUPS

**Imperfect ID Data.** In our experiments, ImageNet-1k and its subset ImageNet-200-hard serve as imperfect ID data. The ImageNet-1k dataset, sourced mainly from web search engines and platforms like Flickr, is large but highly variable in quality Deng et al. (2009). A significant proportion of the images contain ID objects whose features corresponding to their assigned class labels occupy only about half of the image area Kisel et al. (2024); Luccioni & Rolnick (2023); Paullada et al. (2021), rendering the dataset notably imperfect. For ImageNet-200-hard, we further excluded samples that contain only features corresponding to the label from original version, retaining only those samples where the features related to the label constitute only a portion of the image, making it a more challenging imperfect ID dataset. Please refer to Appendix C for more details.

**OOD Datasets.** To evaluate performance in the imperfect ID setting, we utilize the bounding box annotations from the ImageNet validation dataset to mask the ID features relevant to the class labels, retaining only the OOD features. This dataset, referred to as the ImageNet-Real-O, served as the OOD test data in the imperfect ID data setting. Furthermore, we utilize four common benchmarks as OOD test datasets for the unseen OOD setup: iNaturalist Horn et al. (2018), Textures Cimpoi et al. (2014), Places365 Zhou et al. (2018) and SUN Xu et al. (2015). Additionally, Oxford-IIIT Pet Parkhi et al. (2012), SSB-hard Vaze et al. (2022) and NINCO Bitterwolf et al. (2023) serve as the OOD test datasets for the near OOD setup. For the counterpart methods that use additional auxiliary OOD data to fine-tune the model, we employ ImageNet-21k-Resize Ridnik et al. (2021) as the auxiliary OOD dataset, from which we remove the semantically overlapping portion with ImageNet-1k.

**Baseline Methods.** We compare our method DIID with several post-hoc methods that rely solely on ID data, including MSP Hendrycks & Gimpel (2017), Energy Liu et al. (2020), MAHA Lee et al. (2018c), MaxLogit Hendrycks et al. (2022), KNN Sun et al. (2022), GradNorm Huang et al. (2021), ReAct Sun et al. (2021), ASH Djurisic et al. (2022), and NPOS Tao et al. (2023). Additionally, to further investigate the performance of our method, we compared it with Outlier Exposure (OE) Hendrycks et al. (2019), Energy (w/OE) Liu et al. (2020), ATOM (Chen et al., 2021b), POEM (Ming et al., 2022), DOE (Wang et al., 2023b), and DAL (Wang et al., 2023a), which fine-tune the model using additional auxiliary OOD data and typically achieve better OOD detection performance than methods that rely solely on ID data.

**Implementation Details.** We utilize different backbone networks for ImageNet-1k and ImageNet-200-hard as imperfect ID data. For ImageNet-1k, ResNet-50 He et al. (2016), Wide-ResNet-50-2 Zagoruyko & Komodakis (2016), MobileNetV2 Sandler et al. (2018), and ViT-B/16 Dosovitskiy et al. (2020) are employed with pre-trained parameters from the PyTorch official release. For ImageNet-200-hard, ResNet-18 He et al. (2016) is adopted, pre-trained for 100 epochs using cross-entropy loss. The optimization is conducted via SGD with momentum 0.9, weight decay 0.0005, and an initial learning rate of 0.1 with the cosine annealing decay schedule Loshchilov & Hutter (2017).

**Evaluation Metrics.** We evaluate OOD detection performance using three standard metrics: (a) FPR95: the false positive rate for OOD data when the true positive rate for ID data reaches 95%; (b) AUROC: the area under the receiver operating characteristic curve; and (c) ID ACC.: the classification accuracy of ID data. Higher (↑) AUROC and ID ACC. and lower (↓) FPR95 are expected.

Table 1: Comparison between our method and advanced methods on ImageNet-1K. Values are percentages averaged over 10 runs. Bold font indicates the best results in the column.

| Method | ResNet-50 | | Wide-ResNet-50-2 | | MobileNetV2 | | ViT-B/16 | |
|---|---|---|---|---|---|---|---|---|
| | FPR95↓ | AUROC↑ | FPR95↓ | AUROC↑ | FPR95↓ | AUROC↑ | FPR95↓ | AUROC↑ |
| Using Imperfect ID Data and Auxiliary OOD Data | | | | | | | | |
| OE | 34.79 | 92.40 | 28.95 | 93.44 | 39.35 | 91.80 | 27.86 | 92.75 |
| Energy(w/OE) | 35.61 | 90.83 | 26.80 | 93.17 | 37.85 | 91.96 | 26.95 | 92.75 |
| ATOM | 33.27 | 92.21 | 26.97 | 94.29 | 36.90 | 91.53 | 24.77 | 91.62 |
| POEM | 32.34 | 92.50 | 24.42 | 95.05 | 35.65 | 92.60 | 23.84 | 92.52 |
| DOE | 33.51 | 90.65 | 25.43 | 94.31 | 36.44 | 91.42 | 23.67 | 92.31 |
| DAL | 30.79 | 93.72 | 23.20 | 95.67 | 34.53 | 92.71 | 21.46 | 93.20 |
| Using Imperfect ID Data Only | | | | | | | | |
| MSP | 50.01 | 90.67 | 68.02 | 88.78 | 46.50 | 88.20 | 40.90 | 91.88 |
| Energy | 91.75 | 61.80 | 96.39 | 51.54 | 69.12 | 72.86 | 32.94 | 88.21 |
| MAHA | 91.02 | 50.01 | 60.71 | 79.43 | 98.01 | 49.03 | 94.01 | 51.23 |
| GradNorm | 98.06 | 41.23 | 96.07 | 47.34 | 97.68 | 38.39 | 97.38 | 43.13 |
| NPOS | 76.77 | 79.62 | 29.30 | 87.44 | 64.43 | 81.06 | 73.94 | 80.65 |
| KNN | 92.69 | 49.82 | 26.76 | 89.76 | 93.40 | 54.03 | 95.69 | 25.19 |
| MaxLogit | 38.00 | 87.20 | 45.30 | 83.12 | 49.92 | 85.59 | 28.47 | 90.68 |
| DIID (Ours) | $\mathbf{23.40}_{\pm0.5}$ | $\mathbf{96.28}_{\pm0.3}$ | $\mathbf{13.75}_{\pm0.4}$ | $\mathbf{97.95}_{\pm0.2}$ | $\mathbf{27.29}_{\pm0.3}$ | $\mathbf{95.18}_{\pm0.2}$ | $\mathbf{4.76}_{\pm0.4}$ | $\mathbf{99.04}_{\pm0.5}$ |

**DIID Default Setups.** DIID is run for 5 epochs with a batch size of 128 for imperfect ID data, the threshold for disentangling is set to $\tau = 0.6$, the trade-off parameters are set to $\beta = 1.0$ and $\gamma = 5.0$, and the initial learning rate is 0.0001 with cosine decay (Loshchilov & Hutter, 2017).

### 4.2 MAIN RESULTS

Our experimental results primarily investigate the following questions: **(a)** Can our proposed method DIID reliably detect OOD data in scenarios where commonly used ID and OOD datasets exhibit substantial covariate shift? **(b)** Does DIID maintain reliable OOD detection performance in scenarios without covariate shift between ID and OOD data? **(c)** Can DIID preserve robust OOD detection capability in more challenging scenarios where ID and OOD data are semantically similar? Additionally, we provide further results in the appendix, including sensitivity analyses of the hyperparameters $\beta$ and $\gamma$, and more detailed results on ImageNet-200-hard.

**For the first question, detecting OOD features with significant covariate shifts.** We also evaluate our DIID on the unseen OOD setup, where the OOD features rarely appear in the imperfect ID data, thus potentially exhibiting both distinct semantic shifts and substantial covariate shifts relative to the ID features. The experimental results are summarized in Fig. 4, where ImageNet-1k serves as the imperfect ID data, and the results are averaged over four different test OOD datasets. As can be seen, our method DIID still achieves superior OOD detection performance compared to all baseline methods. This demonstrates that DIID not only performs well in the CSFS setting but also exhibits robust OOD detection capabilities across different OOD setups, indicating that DIID is effective not only when the covariate shift between ID and OOD features is small but also when it is large. Please refer to the Appendix D for more detailed results.

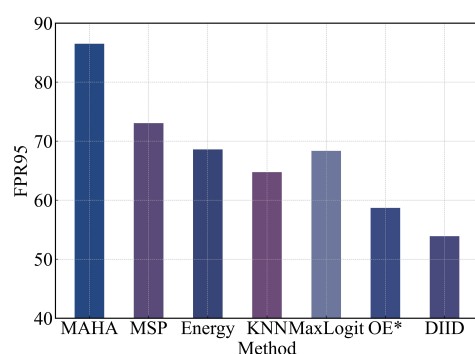

Figure 4: The experimental results of unseen OOD setup with ImageNet-1k as imperfect ID data. * indicates the method need additional auxiliary OOD data to fine-turn the model.

**For the second question, detecting OOD features with negligible covariate shifts.** The main experimental results are summarized in Table 1, where ImageNet-1k serves as imperfect ID data and ImageNet-Real-O serves as the test OOD data. We mainly highlight the following observations: (a) In the CSFS setting, the negligible covariate shift between ID and OOD features leads to suboptimal

Table 2: Comparison between our method and advanced methods on the hard OOD setup with ImageNet-200-hard as imperfect ID data. The bold font indicates the best results in the column.

| Method | Oxford-IIIT Pet | | SSB-hard | | NINCO | | Average | |
|---|---|---|---|---|---|---|---|---|
| | FPR95 ↓ | AUROC ↑ | FPR95 ↓ | AUROC ↑ | FPR95 ↓ | AUROC ↑ | FPR95 ↓ | AUROC ↑ |
| Using Imperfect ID Data and Auxiliary OOD Data | | | | | | | | |
| OE | 67.51 | 72.56 | 68.91 | 78.81 | 64.70 | 77.20 | 67.04 | 76.19 |
| Energy(w/OE) | 68.90 | 72.70 | 65.20 | 79.62 | 67.61 | 76.14 | 67.24 | 76.15 |
| ATOM | 67.37 | 74.84 | 66.43 | 80.94 | 66.29 | 76.87 | 66.70 | 77.55 |
| POEM | 65.86 | 75.39 | 66.40 | 78.55 | 63.51 | 78.93 | 65.26 | 77.62 |
| DOE | 71.42 | 72.01 | 68.23 | 79.21 | 67.84 | 77.04 | 69.16 | 76.09 |
| DAL | 65.19 | 74.46 | 65.79 | 79.34 | 64.89 | 78.31 | 65.29 | 77.37 |
| Using Imperfect ID Data Only | | | | | | | | |
| MSP | 70.27 | 71.06 | 71.50 | 73.65 | 69.52 | 76.37 | 70.43 | 73.69 |
| Energy | 68.43 | 72.80 | 71.86 | 74.23 | 70.46 | 75.42 | 70.25 | 74.15 |
| MAHA | 87.26 | 69.51 | 84.77 | 70.34 | 76.15 | 72.56 | 82.73 | 70.80 |
| GradNorm | 85.44 | 71.32 | 83.14 | 71.69 | 84.61 | 70.41 | 84.40 | 71.14 |
| NPOS | 68.71 | 72.31 | 69.97 | 74.01 | 70.86 | 75.53 | 69.85 | 73.95 |
| KNN | 75.29 | 67.11 | 72.50 | 73.26 | 74.13 | 69.52 | 73.97 | 69.96 |
| MaxLogit | 69.10 | 73.28 | 70.34 | 74.17 | 68.41 | 76.70 | 69.28 | 74.72 |
| DIID (Ours) | **64.03** | **76.01** | **60.50** | **82.49** | **62.89** | **80.24** | **62.47** | **79.58** |

performance of many classical methods that excel in traditional OOD detection settings, which empirically supports our theoretical analysis. (b) Since the test OOD data is derived from imperfect ID data and the test-time imperfect ID data inherently contains both ID features and OOD features, this poses a greater challenge for OOD detection. Furthermore, due to this feature overlap, distance-based methods exhibit significantly degraded performance. (c) Due to the negligible covariate shift between ID and OOD features at test time and the semantic distribution discrepancy between the auxiliary OOD features used during training and the OOD features encountered at test time, OE-based methods also fail to achieve significant performance improvements despite utilizing additional OOD data for model training. (d) Our method, without utilizing any additional auxiliary OOD data, achieves superior OOD detection performance compared to all baseline methods by finetuning models with OOD features disentangled from the imperfect ID data alone. Moreover, despite the negligible covariate shift between ID and OOD features, our method DIID attains an FPR95 of 4.76% when using ViT-B/16 as the backbone network, which is a 16.7% reduction compared to methods trained with extra auxiliary OOD data, thereby demonstrating its robust performance. (e) Our DIID consistently achieves superior OOD detection performance across four distinct backbone models, thereby demonstrating its robustness to diverse model architectures.

**For the third question, detecting OOD features with near semantic space.** We further evaluate our method in the challenging hard OOD setup, where the OOD features are semantically close to the ID features. For this evaluation, we use ImageNet-200-hard as the imperfect ID dataset. For OE-based methods, Places365 is employed as the auxiliary OOD dataset for training. The experimental results are presented in Table 2. We also highlight the following observations: The similar semantic spaces of ID and OOD features lead to suboptimal performance of both types of methods. Despite this, DIID still outperforms all other methods. This is because our method penalizes the model during finetuning for assigning higher prediction scores to OOD features than to ID features, thereby eliminating the interference of OOD features present in the imperfect ID data on detection and endowing the model with stronger and more robust OOD detection capabilities. Specifically, compared with MaxLogit, which also uses only imperfect ID data, DIID achieves a 6.81% reduction in FPR95.

### 4.3 IN-DEPTH ANALYSIS

**Ablation on the Regularization Term.** To validate the effectiveness of our proposed regularization term, we conducted extensive experiments across multiple model architectures using ImageNet-1k as imperfect ID data and ImageNet-Real-O as OOD data. As shown in Fig. 5(a), fine-tuning the models with the additional regularization term consistently improves OOD detection performance across all architectures, thereby demonstrating the effectiveness of the proposed regularization term.

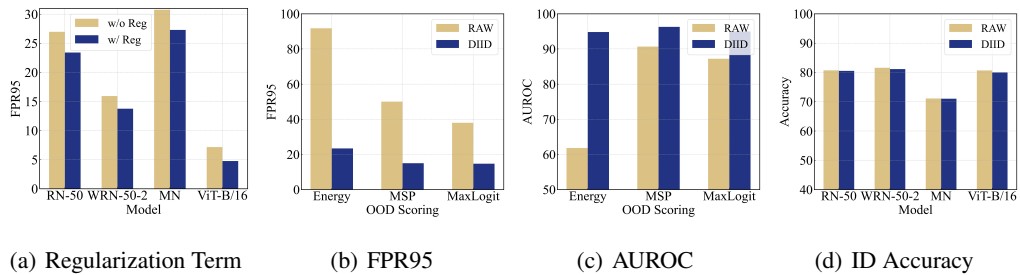

(a) Regularization Term     (b) FPR95     (c) AUROC     (d) ID Accuracy

Figure 5: The experimental results of the ablation study. Fig. 5(a) presents the ablation study results on our proposed regularization term. Figs. 5(b)-5(c) show the OOD detection performance before (RAW) and after (DIID) fine-tuning with our DIID, respectively. Fig. 5(d) shows the classification accuracy on ID data before fine-tuning (RAW) and after fine-tuning with DIID across various backbone models.

**Ablation on the threshold $\tau$.** To explore the effect of different threshold values of $\tau$ on the performance of DIID, we conduct an ablation study, with the experimental results summarized in Fig. 6. We highlight three key observations: (a) Both excessively small and excessively large values of $\tau$ degrade the performance of DIID, as a very small $\tau$ leads to too few disentangled OOD features, while a very large $\tau$ causes the disentangled OOD features to contain ID features, thereby reducing the efficiency of model finetuning; (b) DIID exhibits robust performance across a wide range of $\tau$ values, particularly within [0.4, 0.9], which suggests that meticulous tuning of $\tau$ is not necessary; (c) Although different values of $\tau$ yield varying OOD detection performance, the majority of these values still outperform OE, which relies on additional auxiliary OOD data to fine-tune the model.

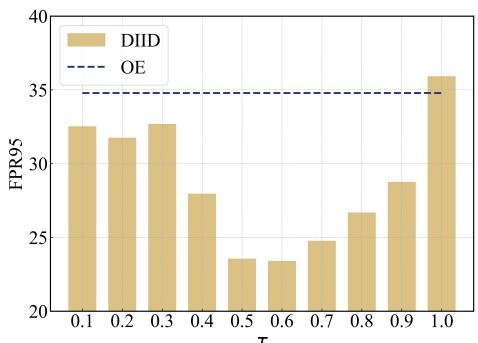

Figure 6: The experimental results of the ablation study about the parameter $\tau$ which is the threshold for disentangling the ID and OOD features contained in the imperfect ID data.

**Ablation on Various OOD Scoring.** To further investigate the performance of our method, we evaluated the OOD detection performance of the same OOD scoring function before and after finetuning the model with DIID. As shown in Figs. 5(b)-5(c), all OOD scoring functions exhibited significantly improved performance after finetuning with DIID. This not only demonstrates the effectiveness of our method but also indicates its agnostic nature with respect to OOD scoring.

**Impact on ID Accuracy.** We also evaluated the classification accuracy on ID data before and after finetuning different models with DIID, using various models as the backbone networks. The experimental results are summarized in Fig. 5(d). As shown, DIID does not compromise the ID classification ability of the model. This indicates that DIID performs well on both tasks of OOD detection: classifying ID data and identifying OOD data that exhibits semantic shifts from ID data.

## 5 CONCLUSION

This paper presents the first study to investigate the impact of covariate shift on OOD detection and provides systematic theoretical proof that significantly vanishing covariate shift degrades the PAC learnability of the OOD detection task. We introduce a more challenging problem, termed Covariate-Shift-Free Setting (CSFS), where the ID and OOD features exhibit negligible covariate shift during testing time. To tackle CSFS, we introduce a novel and efficient method, DIID, which employs class-specific gradients as an effective signal to disentangle OOD features from imperfect ID data for model finetuning, thereby enhancing its OOD detection capability. We conduct extensive experiments, demonstrating consistently superior performance of DIID across diverse OOD detection settings, regardless of whether the covariate shift between ID and OOD features is mild or substantial.

## 6 ETHICS STATEMENT

This study complies with the ICLR Code of Ethics. We propose a novel OOD detection framework and evaluate it on publicly available benchmark datasets. These datasets contain no personally identifiable or sensitive information, thereby ensuring no risks to privacy or security. Our research advances the application of OOD detection in more practical scenarios and holds potential scientific and technological value. All experimental protocols are transparently documented and fairly compared with prior work. The contributions of this study are intended solely for research, supporting the development of artificial intelligence.

## 7 REPRODUCIBILITY STATEMENT

We provide detailed descriptions of our framework, theoretical results, and experimental settings in the paper and appendix. All datasets used are publicly available, and the current description of our method is sufficient for full reproducibility. If the paper is accepted, we will be glad to release the complete implementation to further support the research community.

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

SUPPLEMENTARY MATERIAL: DOES YOUR OOD DETECTOR TRULY DETECT SEMANTICALLY SHIFTED OOD DATA?

## A  RELATED WORK

We review the related studies in OOD detection by categorizing them into two types.

**Utilizing ID Date Only.** The first category of methods relies solely on ID data, without requiring access to OOD samples (Bendale & Boult, 2016; Liu et al., 2024; Sun et al., 2021; Djurisic et al., 2022; Liang et al., 2018a). For example, the baseline MSP (Hendrycks & Gimpel, 2017) estimates confidence using the maximum softmax score. Recent work has explored different aspects of model outputs to design scoring functions, including feature-based methods (Sun et al., 2021; Xu et al., 2024; Liu & Qin, 2024; Zhao et al., 2024; Yuan et al., 2024), logit-based methods (Hendrycks et al., 2022; Wang et al., 2022; Liu et al., 2020), probability-based methods (Sun & Li, 2022; Liang et al., 2018b), and gradient-based methods (Huang et al., 2021). Moreover, distance-based methods such as KNN (Sun et al., 2022) and NPOS (Tao et al., 2023) also utilize features to compute OOD scores, while MAHA (Lee et al., 2018c) employs class-conditional Gaussian distributions to separate OOD and ID data. Additionally, several methods focus on enhancing the discriminative power of activations, such as ASH (Djurisic et al., 2023) and Scale (Xu et al., 2024), which adjust activation patterns for better separation. Graph-based methods, like feature graph analysis (Kim et al., 2023), further refine the detection process by examining the feature structure.

**Utilizing Additional Auxiliary OOD Data.** Another effective approach for OOD detection involves incorporating auxiliary OOD data into the training process (Bitterwolf et al., 2022; Chen et al., 2021a; Sehwag et al., 2021; Mohseni et al., 2020). This strategy often focuses on leveraging low OOD score samples to reduce overfitting and improve detection performance Hendrycks et al. (2019); Liu et al. (2020); Huang et al. (2023). Key techniques include sampling strategies Li & Vasconcelos (2020); Ming et al. (2022), adversarial robust learning Li & Vasconcelos (2020); Lee et al. (2018c); Hein et al. (2019), meta-learning Jeong & Kim (2020), regularization methods Van Amersfoort et al. (2020), contrast learning Lu et al. (2024), and selective sampling Jiang et al. (2024).Additionally, ISH Xu et al. (2024) incorporates scaling directly into training, and Split-Ensemble Chen et al. (2024) partitions the classification task within a single model to better capture OOD signals.When real OOD data is unavailable, some approaches instead focus on outlier synthesis to generate pseudo-OOD samples for improved robustness Lee et al. (2018b); Du et al. (2022); Tao et al. (2023); Wang et al. (2023b); Mirzaei et al. (2024).However, despite these advancements, the gap between auxiliary OOD samples and real-world OOD data remains a significant challenge, similar to the domain adaptation problem where training-test distribution mismatch limits generalization Luo et al. (2023; 2020).

Our method disentangles OOD features from imperfect ID data to fine-tune the model, since the resulting ID and OOD features exhibit no covariate shift, thereby achieving more reliable performance and enhanced robustness under CSFS as well as other diverse scenarios. Moreover, our experimental results show that our method, despite relying solely on imperfect ID data, achieves performance superior to methods that leverage additional auxiliary OOD data, thereby demonstrating its superiority.

## B  THEORY OF LEARNABILITY ON OUR PROPOSED CSFS PROBLEM

### B.1  RISK AND STRONG LEARNABILITY UNDER PAC THEORY

We follow the first learnability framework in (Fang et al., 2022) to study the learnability of our problem. Our main question is whether the learnability of OOD detection decreases with decreasing covariate shift between ID and OOD samples. We let $Z$ denote the domain label, i.e., $Z = 1$ denotes that $\mathbf{x}$ belongs to the OOD class and $Z = 0$ denotes that $\mathbf{x}$ belongs to the ID classes. We first characterize this intuition using formal definition:

**Definition 1** (Covariate Shift). *For OOD samples* $\mathbf{x} \sim D_{X_O}$ *and ID samples* $\mathbf{x} \sim D_{X_I}$, *the covariate shift is quantified by* $T(D_X, D_{X_I})$ *with some pre-defined distributional divergence* $T$.

We choose $T$ as the commonly adopted Total-Variation (TV) divergence (Cover, 1999) in our analysis. Consequently, we have the following conclusion:

$$T(D_{X_O}, D_{X_I}) = \int |pO(\mathbf{x}) - p^I(\mathbf{x})| d\mathbf{x}. \tag{7}$$

Following (Fang et al., 2022), letting ID classes equal to $\{1, 2, \ldots, K\}$ and OOD class equal to $K + 1$ (without generality), we then define two risk for OOD detection as follows:

$$R_D^I(\mathbf{h}) = \mathbb{E}_{(\mathbf{x},y) \sim D_{X_I Y_I}} \ell(\mathbf{h}(\mathbf{x}), y), \tag{8}$$

$$R_D^O(\mathbf{h}) = \mathbb{E}_{(\mathbf{x}) \sim D_{X_O}} \ell(\mathbf{h}(\mathbf{x}), K + 1), \tag{9}$$

where the loss function $l$ satisfies that $l(\mathbf{x}, y) = 0$ if $\mathbf{x} = y$. Following (Chu et al., 2004), we adopt the 0-1 formulation of $l$ as $l(h(\mathbf{x}), y) = 1(h(\mathbf{x}) \neq y)$. Furthermore, we can decompose the risk $R_D^I(\mathbf{h})$ into two components, i.e., the risk of classifying ID samples in ID classes, and the risk of classify ID samples into OOD class $(K + 1)$:

$$\begin{aligned} R_D^I(\mathbf{h}) &= \mathbb{E}_{(\mathbf{x},y) \sim D_{X_I Y_I}} 1(h(\mathbf{x}) \neq y) \\ &= \mathbb{E}_{(\mathbf{x},y) \sim D_{X_I Y_I}} 1(h(\mathbf{x}) \neq y, h(\mathbf{x}) \in 1, 2, \ldots, K) + \mathbb{E}_{(\mathbf{x},y) \sim D_{X_I Y_I}} 1(h(\mathbf{x}) = K + 1). \end{aligned} \tag{10}$$

Since the ratio of ID samples to OOD samples is unknown in prior, the $\alpha$-risk is proposed in (Fang et al., 2022) as a weighted sum of $R_D^I(\mathbf{h})$ and $R_D^O(\mathbf{h})$ to simulate mixed distributions of ID and OOD samples:

**Definition 2** ($\alpha$-Risk).

$$R_D^\alpha = \alpha R_D^I(\mathbf{h}) + (1 - \alpha) R_D^O(\mathbf{h}). \tag{11}$$

Consequently, Fang et al. (2022) defines strong learnability under the risk, i.e., a PAC theory inspired learnability:

**Proposition 1** (Strong Learnability under Risk). *Assuming $D_{XY} = \{D_{XY}^\alpha : \forall \alpha \in [0, 1)\}$, then $\forall \alpha \in [0, 1], \exists \epsilon_{cons}(n), such that \lim_{n \to \infty} \epsilon_{cons}(n) = 0$, we have*

$$\mathbb{E}_{S \sim D_{XY}^{n,I}} [R_D^\alpha A(S) - \inf_{\mathbf{h} \in \mathcal{H}} R_D^\alpha(\mathbf{h})] \leq \epsilon_{cons}(n), \tag{12}$$

*if $\exists \mathbf{h}_n$, such that*

$$R_D^I(\mathbf{h}_n) \to \inf_{\mathbf{h} \in \mathcal{H}} R_D^I(\mathbf{h}_n), \text{ and } R_D^O(\mathbf{h}_n) \to \inf_{\mathbf{h} \in \mathcal{H}} R_D^O(\mathbf{h}_n) \tag{13}$$

*Proof.* According to the condition: $\exists \mathbf{h}_n$, such that

$$R_D^I(\mathbf{h}_n) \to \inf_{\mathbf{h} \in \mathcal{H}} R_D^I(\mathbf{h}_n), \text{ and } R_D^O(\mathbf{h}_n) \to \inf_{\mathbf{h} \in \mathcal{H}} R_D^O(\mathbf{h}_n), \tag{14}$$

one can easily conclude that:

$$\inf_{\mathbf{h} \in \mathcal{H}} R_D^\alpha(\mathbf{h}) = (1 - \alpha) \cdot \inf_{\mathbf{h} \in \mathcal{H}} R_D^I(\mathbf{h}) + \alpha \cdot \inf_{\mathbf{h} \in \mathcal{H}} R_D^O(\mathbf{h}). \tag{15}$$

Then by Theorem 2 (Fang et al., 2022), the strong learnability under risk holds. $\square$

### B.2 BAYESIAN OPTIMAL ERROR AND COVARIATE SHIFT

We then inform that, the optimal error achieved by any hypothesis $h$ on classifying ID and OOD samples, i.e., the Bayesian error, exhibits decrease with decreasing covariate shift:

**Lemma 1.** *Letting $\epsilon = \mathbb{E}_X[\min\{P(Z = 1|X), p(Z = 0|X)\}]$ denote the Bayesian Optimal Error, then we have:*

$$\epsilon = \frac{1}{2} - \frac{1}{2} \cdot T(\pi \cdot P^O, (1 - \pi) \cdot P^I), \tag{16}$$

*where $T$ refers to the TV divergence between $D_{X_I}$ and $D_{X_O}$.*

*Proof.* The Bayesian Optimal Error is:

$$
\begin{aligned}
\epsilon &= \mathbb{E}_X[\min\{P(Z=1|X), p(Z=0|X)\}] \\
&= \mathbb{E}_X[\min\{P(X|Z=1)\frac{P(Z=1)}{P(X)}, p(X|Z=0)\frac{P(Z=0)}{P(X)}\}]dX \\
&= \int \min\{\pi \cdot P^O(X), (1-\pi) \cdot P^I(X)\}\frac{\pi \cdot P^O + (1-\pi) \cdot P^I}{\pi \cdot P^O + (1-\pi) \cdot P^I}dX \\
&= \int \min\{\pi \cdot P^O, (1-\pi) \cdot P^I\}dX \\
&= \frac{1}{2}(\pi \cdot P^O + (1-\pi) \cdot P^I) - |\pi \cdot P^O - (1-\pi) \cdot P^I|dX \\
&= \frac{1}{2} - \frac{1}{2} \cdot T(\pi \cdot P^O - (1-\pi) \cdot P^I)
\end{aligned}
\tag{17}
$$

$\square$

We note that any hypothesis $h$ achieving Bayesian Optimal Error yields the ideal OOD/ID classification performance, i.e., the minimal risk.

### B.3 PROOF OF OUR THEOREM 1

Finally, we are now ready to prove our Theorem 1 presented in the main paper:

*Proof.* We first decompose the Bayesian error $\epsilon$ in below:

$$
\begin{aligned}
\epsilon &= \mathbb{E}_X[\min\{P(Z=1|X), p(Z=0|X)\}] \\
&= \mathbb{E}_{\substack{X \in \Omega_1 \\ X \sim \pi \cdot D_{X_O} + (1-\pi)D_{X_I}}}[P(Z=1|X)] + \mathbb{E}_{\substack{X \in \Omega_2 \\ X \sim \pi \cdot D_{X_O} + (1-\pi)D_{X_I}}}[P(Z=0|X)] \\
&= \pi\mathbb{E}_{\substack{X \in \Omega_1 \\ X \sim D_{X_O}}}[P(Z=1|X)] + (1-\pi)\mathbb{E}_{\substack{X \in \Omega_1 \\ X \sim D_{X_I}}}[P(Z=1|X)] \\
&\quad + \pi\mathbb{E}_{\substack{X \in \Omega_2 \\ X \sim \cdot D_{X_O}}}[P(Z=0|X)] + (1-\pi)\mathbb{E}_{\substack{X \in \Omega_2 \\ X \sim D_{X_I}}}[P(Z=0|X)],
\end{aligned}
\tag{18}
$$

where $\Omega_1$ and $\Omega_2$ follows the definitions in below:

$$
\Omega_1(X) = \{X|P(Z=1|X) \leq P(Z=0|X)\},
\tag{19}
$$

and

$$
\Omega_2(X) = \{X|P(Z=1|X) > P(Z=0|X)\}.
\tag{20}
$$

We then proof the theorem by contradiction. Assuming that for any $\{\mathbf{h}_n\}$ satisfying the condition for proposition 1, we decrease the covariate shift as $T(D_{X_O}, D_{X_I})$. By Lemma 1, we have that the Bayesian optimal error will increase such that the lower bound of the classification error for any $\mathbf{h}$ will increase. Meanwhile, by considering $\ell$ as a 0-1 loss function, we can turn the risk $R_D^I(\mathbf{h}_n)$ and $R_D^O(\mathbf{h}_n)$ with estimated probability $\widehat{P}_{\mathbf{h}_n}(Z=1 \mid X)$ (via $\mathbf{h}_n$) as follows:

$$
R_D^I(\mathbf{h}_n) = R_D^{IDC}(\mathbf{h}_n) + \mathbb{E}_{\mathbf{x} \sim D_{X_I}}[\widehat{P}_{\mathbf{h}_n}(Z=1|X)],
\tag{21}
$$

and

$$
R_D^O(\mathbf{h}_n) = \mathbb{E}_{\mathbf{x} \sim D_{X_O}}[\widehat{P}_{\mathbf{h}_n}(Z=0|X)],
\tag{22}
$$

where $R_D^{IDC}(\mathbf{h}_n)$ refers to the risk of classifying ID sample inside the ID classes. In similar, the risk $R_D^O(\mathbf{h}_n)$ and $R_D^I(\mathbf{h}_n)$ can be further decomposed into regions $\Omega_1$ and $\Omega_2$. Consequently, we discuss the following four situations with increasing $\epsilon$:

- When $\mathbb{E}_{\substack{X \in \Omega_1 \\ X \sim D_{X_O}}} P(Z=1|X)$ increases, $\mathbb{E}_{\substack{X \in \Omega_2 \\ X \sim D_{X_O}}} \widehat{P}_{\mathbf{h}_n}(Z=0|X)$ also increases, and $R_D^O(\mathbf{h}_n)$ increases as well.

- When $\mathbb{E}_{\substack{X \in \Omega_1 \\ X \sim D_{X_{\mathrm{I}}}}} P(Z = 1|X)$ increases, $\mathbb{E}_{\substack{X \in \Omega_1 \\ X \sim D_{X_{\mathrm{I}}}}} \widehat{P}_{\mathbf{h}_n}(Z = 1|X)$ also increases such that $R_D^{\mathrm{I}}(\mathbf{h}_n)$ increases;

- When $\mathbb{E}_{\substack{X \in \Omega_2 \\ X \sim D_{X_{\mathrm{O}}}}} P(Z = 1|X)$ increases, $\mathbb{E}_{\substack{X \in \Omega_1 \\ X \sim D_{X_{\mathrm{I}}}}} \widehat{P}_{\mathbf{h}_n}(Z = 0|X)$ also increases such that $R_D^{\mathrm{O}}(\mathbf{h}_n)$ increases;

- When $\mathbb{E}_{\substack{X \in \Omega_2 \\ X \sim D_{X_{\mathrm{I}}}}} P(Z = 0|X)$ increases, $\mathbb{E}_{\substack{X \in \Omega_1 \\ X \sim D_{X_{\mathrm{I}}}}} \widehat{P}_{\mathbf{h}_n}(Z = 1|X)$ also increases such that $R_D^{\mathrm{I}}(\mathbf{h}_n)$ increases;

Overall, with increasing Bayesian Optimal Error, at least one of $R_D^{\mathrm{I}}(\mathbf{h}_n)$ and $R_D^{\mathrm{O}}(\mathbf{h}_n)$ will enlarge for the fixed $h_n$. Consequently, the simultaneous convergence condition in Proposition 1 tend to fail, as the converged point is the infinum. Therefore, our claim follows.

$\square$

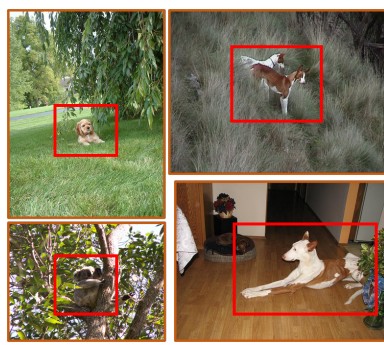 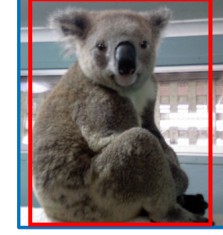 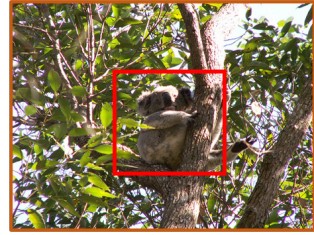

Remove          Remain

(a) Why ImageNet-1k Imperfect             (b) Make ImageNet-200 More Imperfect

Figure 7: Fig. 7(a) presents the data from ImageNet-1K. Fig. 7(b) illustrates the process of constructing ImageNet-200-hard.

## C  ADDITIONAL EXPERIMENTAL DETAILS

**Details About Imperfect ID Data.**

Most of the data in the ImageNet dataset are as shown in Fig. 7(a), where the features related to the labels (enclosed by red boxes) occupy only a small portion of the image. Additionally, due to the broad coverage of ImageNet, the OOD features during testing overlap with those in ImageNet. As a result, when it is used as ID data, the covariate shift between ID and OOD features is minimal. Therefore, it can serve as a large-scale imperfect ID dataset.

Fig. 7(b) illustrates the process of constructing the more challenging ImageNet-200-Hard imperfect ID dataset compared to the original ImageNet-200. It shares the same categories as the ImageNet-200-C dataset (Hendrycks & Dietterich, 2018). We removed samples from the original ImageNet-200 where the features of the target classes occupied a large proportion of the image, retaining only those samples where the target class features constituted a small portion of the image. This modification enhances the challenge posed by the dataset, as ID features account for only a small portion while the majority are OOD features, thereby increasing the difficulty of distinguishing between ID and OOD.

**Details About the Experimental Environment.** The experiments were conducted on a high-performance computing system equipped with 8 NVIDIA RTX 4090 GPUs and an Intel Core i9-14900K CPU. The software environment included Python version 3.10 and CUDA version 12.5.

## D  ADDITIONAL RESULTS

In this section, we provide more detailed experimental results.

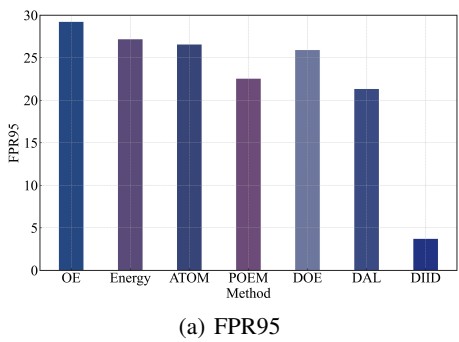 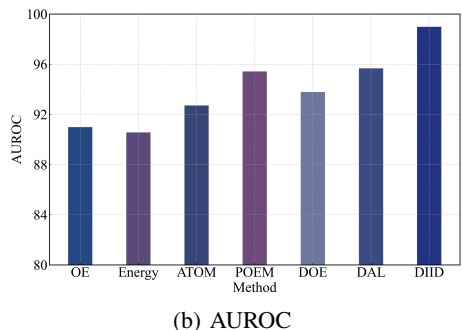

(a) FPR95                    (b) AUROC

Figure 8: Comparison between our method DIID and advanced methods on unseen OOD setup where ImageNet-200-hard serves as imperfect ID data.

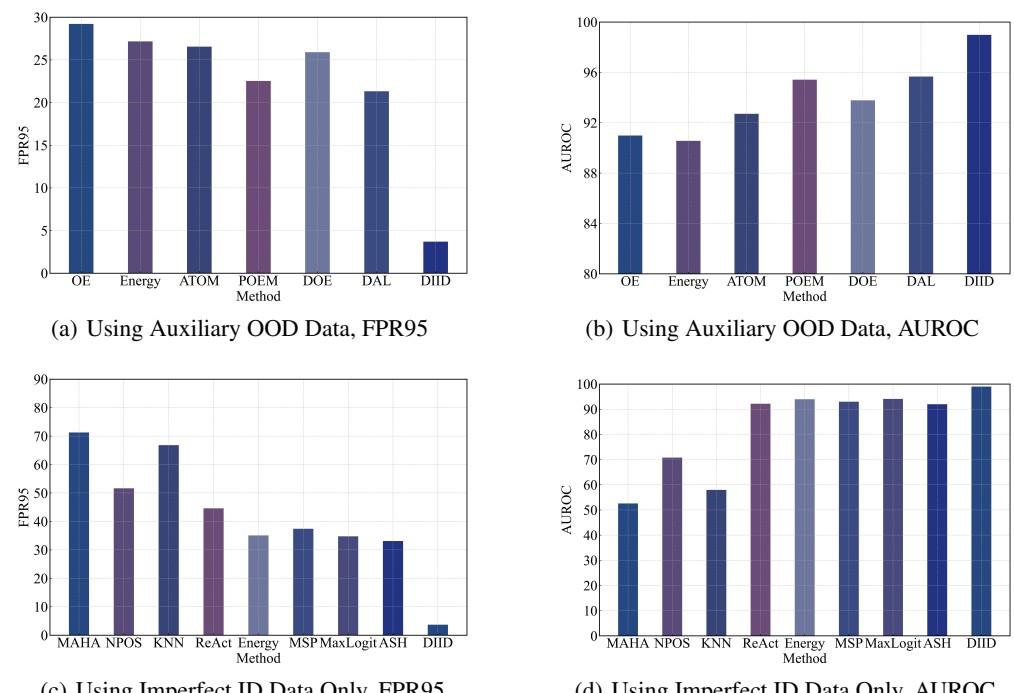

Figure 9: Comparison between our method DIID and advanced methods on ImageNet-200-hard. Figs. 9(a)-9(b) show the results comparing DIID with methods that require additional auxiliary OOD data to fine-tune the model. Figs. 9(c)-9(d) show the results comparing DIID with methods that use only imperfect ID data.

**Imperfect ID Data Setting.** In Fig. 9, we present the experimental results when ImageNet-200-hard is used as the imperfect ID data, with ResNet-18 serving as the backbone network. Consistent with the results provided in the main text for ImageNet-1k, our method not only outperforms methods that solely use imperfect ID data but also achieves better performance than models trained with additional auxiliary OOD data. This further underscores the reliability and robustness of our approach.

**Unseen OOD Setup.** We provide the detailed results on the Unseen OOD setup in Fig. 8, where the imperfect ID data is ImageNet-200-hard. Consistent with the observations highlighted in the main body, our method outperforms all baseline methods. Furthermore, the experimental results demonstrate that DIID maintains strong OOD detection performance even when there is minimal covariate shift between the ID and OOD data, thereby highlighting its reliability and robustness.

**Ablation Study on OOD Scoring.** In addition to the ablation studies of OOD scoring methods presented in the main text, we have conducted further ablation studies across multiple model architectures. The results, as shown in Table 3, demonstrate that our method consistently achieves excellent performance when utilizing various OOD scoring techniques with different models as the backbone networks. This highlights the generalizability and robustness of our approach.

**Ablation Study on Parameter $\tau$.** Here, we provide more detailed experimental results of the ablation studies on the parameter $\tau$ in Table 4. In addition to the FPR95 results presented in the main body, we also report the AUROC and ID ACC metrics. We highlight two key observations: (a) There is a broad range of $\tau$ values that can enhance the performance of our method. Although the performance varies in terms of FPR95, consistent and excellent performance is achieved across all $\tau$ values when evaluated using the more comprehensive AUROC metric. (b) Different values of $\tau$ have a negligible impact on the classification accuracy of the ID data. This indicates that the features decoupled by our method are predominantly OOD features, which further demonstrates the strong ability of our method to decouple OOD features from imperfect ID data.

Table 3: The ablation study on OOD scoring functions across various backbone models, with ImageNet-1k serving as the imperfect ID data. Bold font indicates the best results in the column.

| Model | Method | FPR95↓ | | AUROC↑ | |
|---|---|---|---|---|---|
| | | RAW | DIID | RAW | DIID |
| ResNet-50 | MSP | 50.01 | **23.40** | 90.67 | **96.28** |
| | Energy | 91.75 | **14.98** | 61.80 | **94.82** |
| | MaxLogit | 38.00 | **14.70** | 87.20 | **94.96** |
| Wide-ResNet-50-2 | MSP | 68.02 | **13.75** | 88.78 | **97.95** |
| | Energy | 96.39 | **11.83** | 51.54 | **95.94** |
| | MaxLogit | 45.30 | **10.89** | 83.12 | **96.19** |
| MobileNetV2 | MSP | 46.50 | **27.29** | 88.20 | **95.18** |
| | Energy | 69.12 | **19.47** | 72.86 | **95.42** |
| | MaxLogit | 49.92 | **20.72** | 85.59 | **85.59** |
| ViT-B/16 | MSP | 40.90 | **4.76** | 91.88 | **99.04** |
| | Energy | 32.94 | **7.94** | 88.21 | **98.01** |
| | MaxLogit | 28.47 | **8.36** | 90.68 | **97.97** |

Table 4: The ablation study on the parameter $\tau$, with ImageNet-1k serving as the imperfect ID data.

| Parameter-$\tau$ | 0.1 | 0.2 | 0.3 | 0.4 | 0.5 | 0.6 | 0.7 | 0.8 | 0.9 | 1.0 |
|---|---|---|---|---|---|---|---|---|---|---|
| ID ACC. | 80.16 | 80.32 | 80.39 | 80.29 | 80.62 | 80.51 | 80.69 | 80.37 | 80.08 | 80.17 |
| FPR95 | 32.53 | 31.76 | 32.69 | 27.96 | 23.56 | 23.40 | 24.77 | 26.69 | 28.76 | 35.92 |
| AUROC | 94.74 | 94.78 | 94.44 | 95.61 | 96.70 | 96.28 | 96.02 | 95.97 | 95.28 | 93.81 |

**Sensitivity Analysis of Hyper-parameters $\beta$ and $\gamma$.** We utilize ImageNet-1k as the ID data and ImageNet-Real-O as the test OOD data, using ResNet-50 as the backbone model. We highlight the following observations: (a) The hyperparameters of our method show robust performance across a wide range of values, without

Table 5: Sensitivity Analysis of Hyper-parameters $\beta$ and $\gamma$.

| Parameter Value | $\beta$ | | | $\gamma$ | | |
|---|---|---|---|---|---|---|
| | FPR95↓ | AUROC↑ | ID ACC↑ | FPR95↓ | AUROC↑ | ID ACC↑ |
| 1 | 23.40 | 96.28 | 80.51 | 26.31 | 95.72 | 77.92 |
| 2 | 21.16 | 96.97 | 80.59 | 26.74 | 95.96 | 78.65 |
| 3 | 23.71 | 96.15 | 80.17 | 25.58 | 95.79 | 80.26 |
| 4 | 25.64 | 95.21 | 79.26 | 23.57 | 96.79 | 80.27 |
| 5 | 27.70 | 94.26 | 78.56 | 23.40 | 96.28 | 80.51 |

needing overly fine-tuned selection. (b) Across all parameter setups of $\beta$ and $\gamma$, DIID achieves better OOD detection performance than the state-of-the-art approache (FPR95: 30.79%, AUROC: 93.72%) trained with auxiliary OOD data. (c) As $\beta$ increases, the OOD detection performance improves, while the classification accuracy slightly declines. And, as $\gamma$ increases, both OOD detection performance and classification accuracy improve. This demonstrates that our regularization term benefits both objectives, enabling our method to enhance OOD detection while maintaining ID accuracy.

# E  THE USE OF LARGE LANGUAGE MODELS (LLMS)

The authors declare that large language models (LLMs) were used solely for polishing the writing of this manuscript. No part of the theoretical development, algorithm design, experimental implementation, data analysis, or other research-related tasks involved the use of LLMs.

