# OpenReview forum: "Rethinking the Identification Capability of Out-of-Distribution Detection"
_ICLR.cc/2026/Conference — Submitted to ICLR 2026_

### Official Review · Reviewer_v9Gz · 2025-10-28

**Soundness:** 2
**Presentation:** 2
**Contribution:** 2
**Rating:** 2
**Confidence:** 5

**Summary:**

This paper argue that covariate shift between ID and OOD datasets can act as a "shortcut", artificially inflating performance. To investigate this, they introduce the problem setting Covariate-Shift-Free Setting (CSFS), where OOD detection must be performed with negligible covariate shift, and provide theoretical analysis suggesting that OOD detection becomes harder as covariate shift decreases. To tackle the CSFS, they propose DIID (Disentangling Imperfect ID Data), which uses class-specific gradients to disentangle OOD features from "imperfect" ID data and then uses these disentangled features to fine-tune the model.

**Strengths:**

1. The proposed method without requiring any external OOD data, which is application
2. Experiments are relative comprehensive

**Weaknesses:**

1. The observation that OOD samples with minimal covariate shift are more challenging to detect is widely acknowledged in the community. Existing methods typically exhibit inferior performance in near-OOD scenarios compared to far-OOD settings.
2. The strategy of leveraging background regions as OOD features is not novel. many CLIP-based OOD detection approaches, such as LOCOOP and SCT, have similarly utilized background context to model OOD feature.
3. The CSFS setting is constructed in a way that inherently benefits DIID, making the inflated performance in Table 1.
4. The baseline methods used for comparison are relatively weak, with the most recent ones dating back to 2023.

**Questions:**

N/A

---

### Official Review · Reviewer_zuAA · 2025-10-31

**Soundness:** 3
**Presentation:** 2
**Contribution:** 2
**Rating:** 4
**Confidence:** 4

**Summary:**

This work introduces a novel and challenging problem setting called Covariate-Shift-Free Setting (CSFS) for OOD detection. The authors argue that existing OOD detectors may rely on covariate shift (differences in input distributions) as a shortcut instead of truly detecting semantic shift (differences in class labels). To address this, they propose a setting where ID and OOD data have minimal covariate shift, forcing models to focus on semantic differences. It also provides a theoretical analysis showing that OOD detection becomes more difficult as covariate shift decreases. To tackle CSFS, this work proposes DIID, a method that utilizes class-specific gradients to disentangle OOD features from imperfect ID data. It fine-tunes the model using these self-extracted OOD features. Besides, it introduces a regularization term to penalize misclassification of ID data as OOD. Additionally, extensive experiments across multiple benchmarks and model architectures show that DIID outperforms existing methods, even those that use additional OOD data during training.

**Strengths:**

1. The introduction of CSFS addresses a critical oversight in OOD detection literature. The focus on semantic shift vs. covariate shift is well-motivated and important for real-world applications.

2. This work also presents a PAC-learning-based analysis showing that OOD detection becomes harder with decreasing covariate shift. It also links Bayesian optimal error to covariate shift, giving a formal foundation to the claims.

Innovative Methodology:

3. To address the issure, it utilizes class-specific gradients to disentangle OOD features, which is computationally efficient.

4. Empirical results demonstrates that DIID outperforms all baselines across multiple settings (unseen OOD, near OOD, CSFS). It also achieves significant improvements even without external OOD data, which is a major practical advantage.

5. Comprehensive evaluation validates the effectiveness of the propsed approach in iterms of using different backbone and datasets. It also conducated sufficient study on hyperparameters, regularization, and OOD scoring functions.

**Weaknesses:**

1. The use of gradients to identify OOD features is heuristic to some extend and lacks deep theoretical justification. Besides, the choice of the threshold $\tau$ is empirical; a more principled analysis would strengthen the method.

2. Computational overhead: computing input gradients for all training samples during fine-tuning adds non-trivial computational cost, which is not thoroughly discussed.

3. Dependence on initial model: the gradient-based disentanglement assumes the model is already reasonably accurate on ID data. Performance may degrade with poor initial models.

4. Comparison to gradient-based baselines: limited comparison to other gradient-based OOD methods (e.g., GradNorm, GEOM). A deeper comparison would better contextualize the contribution.

5. Theoretical assumptions: the PAC analysis assumes the hypothesis class is well-specified and that the model can achieve optimal performance, which may not hold in practice.

**Questions:**

Please see the **Weakness**

---

### Official Review · Reviewer_Rbnp · 2025-11-02

**Soundness:** 2
**Presentation:** 2
**Contribution:** 2
**Rating:** 4
**Confidence:** 4

**Summary:**

This paper argues that traditional out-of-distribution (OOD) detectors succeed by "cheating"—they detect easy stylistic covariate shifts (like blur or different backgrounds) rather than the true, harder semantic shifts (new, unseen classes). The authors introduce a new, more challenging problem called the Covariate-Shift-Free Setting (CSFS)  to eliminate this "shortcut." They then propose DIID, a novel fine-tuning method that cleverly creates its own OOD data; it uses class-specific gradients to "disentangle" and extract OOD background features from "imperfect" in-distribution (ID) images. This self-generated OOD data is then used to fine-tune the model, which, according to the paper, allows DIID to dramatically outperform all baselines in the new CSFS setting

**Strengths:**

- Compelling Problem Motivation: The paper's primary strength is its clear and intuitive motivation. It argues that previous out-of-distribution (OOD) detection research has a fundamental flaw: detectors may not be learning to identify true semantic shifts (new classes). Instead, they are likely "cheating" by detecting an easier "shortcut"—a covariate shift (stylistic or superficial changes). This central idea, questioning whether detectors are truly "identifying" the right thing, is a valuable and important contribution to the field

**Weaknesses:**

- Lacks Rigorous Mathematical Formulation: The paper's most critical weakness is that its entire premise is built on a qualitative distinction. While it convincingly describes the difference between semantic and covariate shifts, it fails to provide any mathematical formulation to formally separate them. This makes the core problem ill-defined. Ad-Hoc Problem Setting (CSFS): Because the shifts are not formally defined, the paper's novel Covariate-Shift-Free Setting (CSFS) is not a rigorously defined problem. It is an ad-hoc, empirically constructed setting. The authors claim their new ImageNet-Real-O test set (created by masking out ID objects ) is free of covariate shift, but this is an assertion based on experimental design, not a mathematical guarantee.

 - Weak Theoretical Support (Theorem 1): The paper's theoretical claim—that OOD learnability degrades as covariate shift decreases —is not as valuable as it appears. The proof (Appendix B) relies on a total measure of distributional divergence that mixes both semantic and covariate shifts. Since the paper never mathematically separates these two concepts, the theorem cannot actually make a rigorous claim about the effect of only the covariate shift. It merely restates the known fact that as total distribution overlap increases, detection becomes harder.

- Unfair Experimental Comparisons: The paper compares its proposed DIID method against baselines in a misleading way. DIID is explicitly a fine-tuning method that requires re-training the model. The methodology section details its "Learning Strategy" and the experiments state it is "run for 5 epochs". However, it is benchmarked directly against methods the paper itself labels as "post-hoc" (e.g., MSP, Energy, MaxLogit, KNN, GradNorm), which are by definition training-free. The method should be compared against state-of-the-art training-based OOD detection methods such as OE and POEM

**Questions:**

Please refere to the above weakneess

---

### Official Review · Reviewer_iegY · 2025-11-03

**Soundness:** 2
**Presentation:** 2
**Contribution:** 2
**Rating:** 2
**Confidence:** 4

**Summary:**

The paper introduces Covariate-Shift-Free Setting (CSFS) to OOD detection. The paper argues the existing performance of OOD detection deteriorates when covariate shift is introduced. To tackle the proposed CSFS problem, the paper introduces a method DIID, which disentangle ID and OOD features from imperfect OOD data using class-specific gradient. Using ImageNet-1K and subset of ImageNet-200-hard as ID and their proposed ImageNet-Real-O as OOD data, their method outperforms existing methods.

**Strengths:**

1. To the best of my knowledge, DIID is the first method that uses class-specific gradients of a classifier to disentangle ID features from OOD features.
2. On the standard OOD datasets (SSB-hard and NINCO), the proposed DIID method outperform existing methods.
3. The proposed fine-tuning method does not significantly harm the ID accuracy.

**Weaknesses:**

1. This paper fails to point out many previous work that address the covariate-shift problem in OOD detection. Some examples: full-spectrum OOD detection introduces covariate shift datasets into the evaluation of OOD detection algorithms [1], ImageNet-OOD argues that OOD detection algorithms are detecting covariate shift and introduces the ImageNet-OOD dataset to evaluate OOD detection algorithms with minimal influence from covariate shift [2], Zhang et.al found that improvement in OOD detection hurts classifier robustness in covariate shifts [3], IS-OOD is a benchmark proposed with varying degrees of covariate and semantic shift [4], Li et.al argues that OOD detection algorithm cannot and should not detect semantic shift [5]. Given the context of many previous work, the introduction of CSFS isn't novel.
2. The main result in Table 1, which uses a newly introduced ImageNet-Real-O, which uses bounding box of ImageNet to mask its ID features. This dataset construction seems like circular since the DIID method implicitly extract out the bounding box. The resulting performance is also bizarre: Energy and Mahalanobis Distance performing close to random chance, which is atypical; huge gap between Energy and MaxLogit; these methods are typically highly correlated. The overall poor performance on Energy is concerning as this is a very well established method. Deeper analysis on why this occurs is needed to properly justify the evaluation setup.
3. Lack of comparisons using the standard OOD evaluation makes it difficult to place this work in the existing body of literature: model is trained with additional covariate shift data. One potential concern is the added imperfect added is introducing noise into the ID data and the proposed DIID method is simply denoising the data (which would still be useful but not in the context of OOD detection). More analysis is needed to justify this change in the evaluation.
4. More detail is needed to justify the problem formulation. First, defining the foreground (class-relevant features) as ID and background (class-irrelevant features) as OOD is bizarre. Whether a datapoint comes from a distribution p1(x) or different distribution p2(x) is not fully specified by the class. It is just classification models tends to do better at performing this task than generative models [6]. More justification is needed for this formulation as it is not straight-forward. Second, CSFS assumes that OOD features exhibits **no covariate shift** compared to ID features. This formulation seems to create an impossible task. An OOD detection algorithms takes in some feature x as input. If the input distribution are the same, that is P_o(x) = P_i(x), then it is impossible to expect an OOD detection algorithm to differentiate between the two.
5. Minor: typo in utimate line 048. Type in softmax in equation 1. Citation formatting needs to be changed (\citep instead of \cite)

[1] Yang, Jingkang, Kaiyang Zhou, and Ziwei Liu. "Full-spectrum out-of-distribution detection." International Journal of Computer Vision 131.10 (2023): 2607-2622.
[2] Yang, William, Byron Zhang, and Olga Russakovsky. "ImageNet-OOD: Deciphering Modern Out-of-Distribution Detection Algorithms." The Twelfth International Conference on Learning Representations.
[3] Zhang, Qingyang, et al. "The best of both worlds: On the dilemma of out-of-distribution detection." Advances in Neural Information Processing Systems 37 (2024): 69716-69746.
[4] Long, Xingming, et al. "Rethinking the Evaluation of Out-of-Distribution Detection: A Sorites Paradox." Advances in Neural Information Processing Systems 37 (2024): 89806-89833.
[5] Li, Yucen Lily, et al. "Out-of-Distribution Detection Methods Answer the Wrong Questions." ICML 2025.
[6] Kirichenko, Polina, Pavel Izmailov, and Andrew G. Wilson. "Why normalizing flows fail to detect out-of-distribution data." Advances in neural information processing systems 33 (2020): 20578-20589.

**Questions:**

1. Since DIID uses the classifier's gradient to generate a proxy OOD dataset to train on, how does it perform against methods that uses class-specific gradients like ODIN [1]?
2. How does DIID perform if imperfect ID data were excluded?


[1] Liang, Shiyu, Yixuan Li, and R. Srikant. "Enhancing The Reliability of Out-of-distribution Image Detection in Neural Networks." International Conference on Learning Representations. 2018.

---

### Meta-Review · Area_Chair_scuT · 2026-01-04

**Summary:**

The paper received an average rating of 3.0. Specifically, all reviewers initially leaned towards rejection in their original reviews (2, 2, 4, 4). The authors did not provide a rebuttal to address the reviewers' valuable questions and concerns. Thus, the AC recommended a rejection.

**Reviewer Concerns:**

No rebuttal is provided.

**Reviewer Scores:**

No rebuttal is provided.

---

### Decision · Program_Chairs · 2026-01-26

Reject